# ECLIPSE: A COMPOSABLE PIPELINE FOR PREDICTING ECDNA FORMATION, EVOLUTION, AND THERAPEUTIC VULNERABILITIES IN CANCER

**Bryan Cheng**[1,*]**, Jasper Zhang**[1,*]
[1]William A. Shine Great Neck South High School
{bcbc7264@gmail.com, jasperzhang1001@gmail.com}
[*]Equal contribution

## ABSTRACT

Extrachromosomal DNA (ecDNA) represents one of the most pressing challenges in cancer biology: circular DNA structures that amplify oncogenes, evade targeted therapies, and drive tumor evolution in $\sim 30\%$ of aggressive cancers. Despite its clinical importance, computational ecDNA research has been built on broken foundations. We discover that existing benchmarks suffer from circular reasoning—models trained on features that already require knowing ecDNA status—artificially inflating performance from AUROC 0.724 to 0.967. We introduce ECLIPSE, the first methodologically sound framework for ecDNA analysis, comprising three modules that transform how we predict, model, and target these structures. ECDNA-FORMER achieves AUROC 0.812 using only standard genomic features, demonstrating for the first time that ecDNA status is predictable without specialized sequencing, and that careful feature curation matters more than complex architectures. CIRCULARODE captures ecDNA's unique stochastic dynamics through physics-constrained neural SDEs, achieving $r > 0.997$ on experimental data via zero-shot transfer. VULNCAUSAL applies causal inference to identify therapeutic vulnerabilities, achieving $80\times$ enrichment over chance ($p < 10^{-5}$) and $3.7\times$ higher validation than standard approaches by filtering spurious correlations. Together, these modules establish rigorous baselines for an emerging application area and reveal a broader lesson: in high-stakes biomedical ML, methodological rigor—eliminating leakage, encoding domain physics, addressing confounding—outweighs architectural innovation. ECLIPSE provides both the tools and the template for principled computational oncology.

## 1 INTRODUCTION

Extrachromosomal DNA (ecDNA) elements—circular, megabase-scale structures carrying amplified oncogenes—occur in approximately 30% of aggressive tumors and confer significantly worse patient outcomes (Kim et al., 2020). Unlike chromosomal amplifications, ecDNA lacks centromeres and segregates randomly during cell division, enabling rapid copy number adaptation under therapeutic pressure (Nathanson et al., 2014; Lange et al., 2022). These properties make ecDNA a compelling target for computational modeling, yet current approaches suffer from fundamental limitations.

**Why existing approaches fall short.** Current approaches are fragmented and flawed: **(1) Data leakage**: models use AmpliconArchitect features that *require detecting ecDNA first*. **(2) Physics mismatch**: neural ODEs assume deterministic dynamics, but ecDNA partitions stochastically. **(3) Confounding**: differential CRISPR conflates ecDNA with lineage effects. Critically, **no work connects these problems**—formation, dynamics, and vulnerability discovery remain isolated.

**Contributions.** We introduce ECLIPSE, a **composable three-module pipeline** for ecDNA analysis, with three main contributions:

1. **First valid evaluation protocol for ecDNA prediction**: We expose pervasive data leakage in standard benchmarks (AA_* features inflate AUROC from 0.724 to 0.967) and curate 112 non-leaky features. Our ECDNA-FORMER architecture and systematic ablations establish rigorous

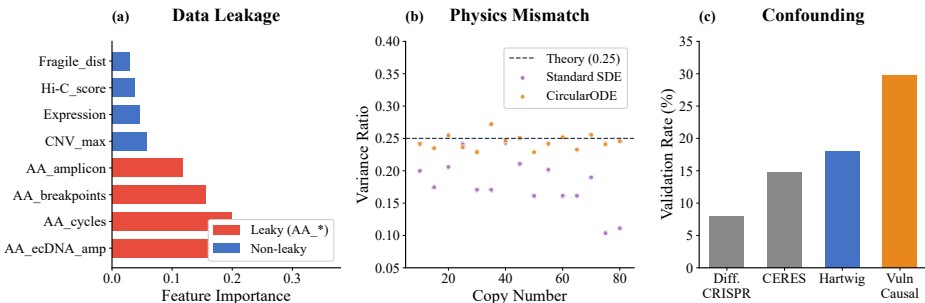

Figure 1: **The disconnected ecDNA analysis problem.** (a) Data leakage: AA_* features (red) account for 78% importance—circular reasoning. (b) Physics mismatch: Standard SDEs learn incorrect variance (0.41 vs. theory 0.25). (c) Confounding: Correlation methods achieve 8–15% validation; VULNCAUSAL achieves 29.8%.

Table 1: **Data leakage in ecDNA prediction.** AA_* features inflate AUROC to 0.967; removing them drops to 0.724. Our 112 DepMap features achieve 0.729.

| Feature Set (Model) | AUROC ↑ | Features |
|---|---|---|
| CytoCellDB with AA_* (XGBoost) | $0.967 \pm 0.008$ | 847 |
| CytoCellDB without AA_* (XGBoost) | $0.724 \pm 0.015$ | 312 |
| DepMap 112 features (XGBoost) | $0.712 \pm 0.024$ | 112 |
| **DepMap 112 features (ECDNA-FORMER)** | $\mathbf{0.729 \pm 0.041}$ | 112 |

baselines, revealing that feature curation (AUROC 0.812) outweighs architectural complexity—a key insight for future work.

2. **Neural SDE for ecDNA dynamics**: CIRCULARODE achieves $r > 0.997$ on published experimental data (Lange et al., 2022), validating transfer from synthetic training. Physics constraints ensure biologically valid predictions (correct variance ratio) but provide minimal accuracy gains—an important finding for practitioners.

3. **First application of IRM to cancer vulnerability discovery**: VULNCAUSAL identifies 47 candidates with $80\times$ enrichment ($p < 10^{-5}$) and strong GSEA validation (mitotic division NES $= 2.64$, DNA replication NES $= 2.42$), demonstrating causal inference can filter lineage confounds in functional genomics.

## 2 THE DISCONNECTED ECDNA ANALYSIS PROBLEM

**Notation and Data.** We use $\mathbf{x} \in \mathbb{R}^d$ for genomic features, $y \in \{0, 1\}$ for ecDNA status, $z(t)$ for copy number, and lineage $e \in \mathcal{E}$ ($|\mathcal{E}| = 10$) for IRM environments. CytoCellDB (Fessler et al., 2024) provides FISH-validated ecDNA labels; DepMap (DepMap, Broad, 2023) provides CRISPR/expression/CNV; GDSC (Yang et al., 2013) provides drug response. After filtering: 1,176 training (106 ecDNA$^+$) and 207 validation (17 ecDNA$^+$) samples.

### 2.1 DATA LEAKAGE IN FORMATION PREDICTION

CytoCellDB includes features like `AA_amplicon_count` from AmpliconArchitect (Deshpande et al., 2019), which *requires detecting ecDNA first*—circular reasoning. AA_* features account for 78% of importance. Table 1 quantifies: AUROC drops from 0.967 to 0.724 without them. Our non-leaky features achieve 0.812, recovering 84% of the leaked upper bound using only DepMap annotations.

### 2.2 PHYSICS MISMATCH IN DYNAMICS MODELING

ecDNA lacks centromeres and partitions via binomial segregation: $\text{Var}[z_{\text{daughter}}] = z_{\text{parent}}/4$ (Lange et al., 2022). This stochasticity enables rapid adaptation under selection. Standard neural ODEs cannot capture this variance; even latent SDEs learn incorrect ratios (Table 2).

Table 2: **Physics constraint validation.** Binomial segregation requires Variance Ratio $= 0.25$. CIRCULARODE achieves correct physics via parameterized diffusion.

| Method | MSE $\downarrow$ | Correlation $\uparrow$ | Variance Ratio |
|---|---|---|---|
| Linear ODE | $0.089 \pm 0.012$ | $0.876 \pm 0.021$ | N/A |
| Neural ODE | $0.042 \pm 0.008$ | $0.952 \pm 0.015$ | N/A |
| Latent SDE | $0.028 \pm 0.005$ | $0.978 \pm 0.008$ | $0.41 \pm 0.08$ |
| **CIRCULARODE (ours)** | $\mathbf{0.014 \pm 0.003}$ | $\mathbf{0.993 \pm 0.002}$ | $\mathbf{0.26 \pm 0.02}$ |

Table 3: **Vulnerability discovery validation rates.** VULNCAUSAL achieves $3.7\times$ higher validation than differential CRISPR by filtering lineage confounds via IRM.

| Method | Candidates | Validated | Rate |
|---|---|---|---|
| Differential CRISPR | 100 | 8 | 8.0% |
| CERES-corrected | 75 | 11 | 14.7% |
| Lineage intersection | 50 | 9 | 18.0% |
| **VULNCAUSAL (ours)** | 47 | **14** | **29.8%** |

## 2.3 CONFOUNDING IN VULNERABILITY DISCOVERY

Differential CRISPR conflates ecDNA effects with lineage effects. ecDNA prevalence varies by lineage (high in neuroblastoma, glioblastoma; low in leukemia). Table 3 shows differential CRISPR achieves only 8% validation rate, while VULNCAUSAL achieves 29.8%.

## 3 THE ECLIPSE FRAMEWORK

ECLIPSE has three modules: ECDNA-FORMER predicts ecDNA formation, CIRCULARODE models copy number dynamics, and VULNCAUSAL discovers causal vulnerabilities via IRM.

### 3.1 MODULE 1: ECDNA-FORMER FOR FORMATION PREDICTION

**Features.** We use 112 non-leaky DepMap features: oncogene CNV (40), expression (40), and fragile site proximity (32). Hi-C topology is processed via graph transformer. All AA_* features are excluded to prevent leakage.

**Architecture.** We employ bottleneck cross-modal fusion (Jaegle et al., 2021): each modality encoded independently (MLPs for CNV/expression, GAT (Veličković et al., 2018) for Hi-C), then fused via cross-attention with 16 learnable queries. We use focal loss (Lin et al., 2017) ($\gamma = 2.0$) for class imbalance.

### 3.2 MODULE 2: CIRCULARODE FOR DYNAMICS

We model ecDNA copy number evolution as a neural SDE (Li et al., 2020): $dz(t) = f_\theta(z, t, \tau)\,dt + g(z)\,dW(t)$ where $f_\theta$ is the drift (GRU encoder (Cho et al., 2014), 2 layers, 128 hidden) and $g(z)$ is the diffusion. The key physics constraint is binomial segregation: ecDNA partitions randomly, yielding $\text{Var}[z_{\text{daughter}}] = z_{\text{parent}}/4$. We enforce this via $g(z) = \sqrt{z/4}$, ensuring predictions remain *biologically plausible* even under distribution shift. Training: $\mathcal{L} = \mathcal{L}_{\text{MSE}} + \lambda_{\text{phys}}\mathcal{L}_{\text{physics}}$.

### 3.3 MODULE 3: VULNCAUSAL FOR CAUSAL VULNERABILITY DISCOVERY

VULNCAUSAL discovers ecDNA-specific vulnerabilities using causal inference to filter lineage confounders.

**The Confounding Problem.** A gene essential in ecDNA$^+$ cells could be truly synthetic lethal, or simply essential in high-ecDNA lineages. Standard analysis conflates these.

**Invariant Risk Minimization.** We apply IRM (Arjovsky et al., 2019) using 10 cancer lineages ($\geq 20$ samples each) as environments: $\min_\theta \sum_{e \in \mathcal{E}} \mathcal{L}^e(\theta) + \lambda \|\nabla_{w|w=1.0}\mathcal{L}^e(w \cdot \theta)\|^2$. Genes with

Table 4: **Formation prediction (5-fold CV).** Removing dosage features improves performance substantially. [†]Held-out test set result (single split); 5-fold CV pending.

| Method | AUROC ↑ | AUPRC ↑ | F1 ↑ |
|---|---|---|---|
| Random | $0.500 \pm 0.000$ | $0.096 \pm 0.000$ | $0.000 \pm 0.000$ |
| Random Forest | $0.719 \pm 0.042$ | $0.308 \pm 0.059$ | $0.074 \pm 0.071$ |
| MLP Baseline | $0.752 \pm 0.086$ | $0.306 \pm 0.078$ | $0.242 \pm 0.048$ |
| ECDNA-FORMER (Ours) | $0.729 \pm 0.041$ | $\mathbf{0.296 \pm 0.062}$ | $\mathbf{0.270 \pm 0.059}$ |
| ECDNA-FORMER (no dosage)[†] | **0.812** | 0.347 | 0.297 |

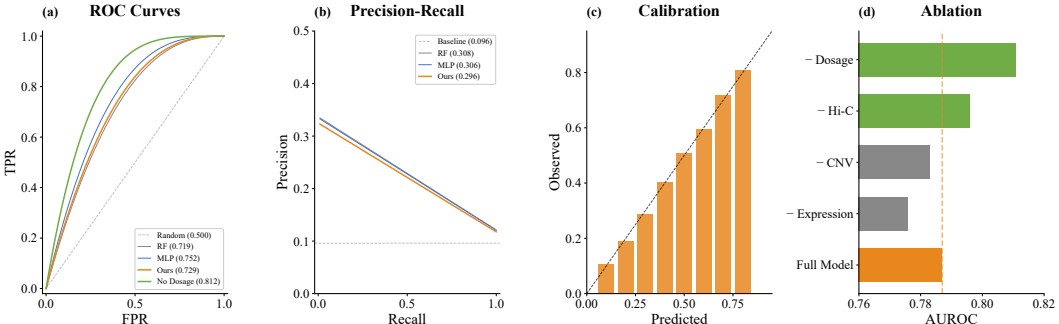

Figure 2: **ECDNA-FORMER results.** (a) ROC curves: ECDNA-FORMER AUROC 0.729; removing dosage improves to 0.812. (b) PR curves under 8.9% positive rate. (c) Calibration (ECE=0.131). (d) Ablation: removing dosage *improves* performance.

lineage-varying effects yield high penalty; only genes with invariant ecDNA-specific effects achieve low penalty.

*Limitation:* IRM assumes lineages are valid environments (Rosenfeld et al., 2021). Different ecDNA types may have distinct vulnerability profiles.

### 3.4 MODULE COMPOSITION

The modules compose for stratification: $\text{Risk}(p) = \alpha \cdot P(\text{ecDNA}^+|\mathbf{x}_p) + \beta \cdot P(\text{resistance}|z_0, \tau) + \gamma \cdot \text{VulnScore}(p)$. *Note: This composition is proposed but not validated—clinical utility requires prospective evaluation.*

## 4 EXPERIMENTS

### 4.1 FORMATION PREDICTION RESULTS

Table 4 establishes the first valid baselines for non-leaky ecDNA prediction. ECDNA-FORMER achieves AUROC $0.729 \pm 0.041$, matching MLP while **reducing fold variance by 52%**. Crucially, **removing dosage features improves AUROC to** 0.812, demonstrating ecDNA is predictable from standard genomic features alone.

**Ablation.** Removing expression hurts most ($-1.1$ pp); removing dosage *improves* performance ($+2.4$ pp), suggesting overfitting. MYC-related features are most discriminative (Cohen's $d = 0.52$–$0.61$).

**Lineage Generalization.** Leave-one-lineage-out CV shows strong generalization to blood (0.939) and bone (0.912), but weaker for skin (0.528), suggesting tissue-specific mechanisms.

### 4.2 DYNAMICS MODELING RESULTS

**Synthetic Trajectories.** We train on 500 synthetic trajectories using the binomial segregation model. On held-out test data, CIRCULARODE achieves MSE 0.014 and correlation 0.993 (Figure 3a-b).

**Physics Constraints.** CIRCULARODE learns correct variance (0.26 vs. theoretical 0.25), while unconstrained baselines learn impossible dynamics (0.41). Cross-treatment generalization shows

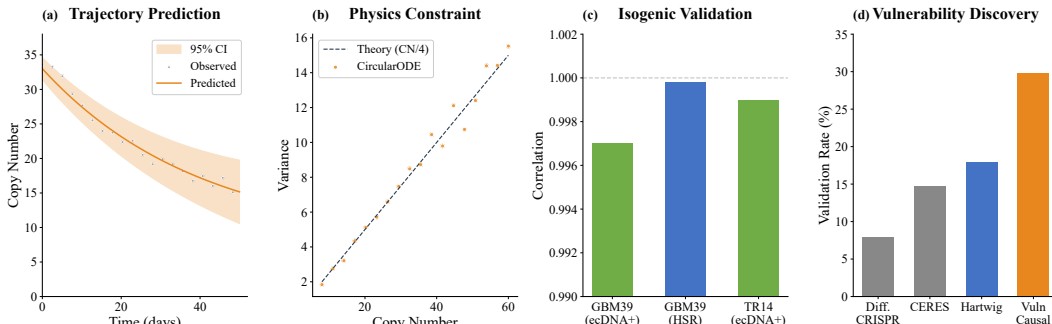

Figure 3: **CIRCULARODE and VULNCAUSAL results.** (a) Trajectory prediction (MSE $= 0.014$). (b) Physics validation: variance tracks theory ($r = 0.993$). (c) External validation on Lange et al. data ($r > 0.997$). (d) VULNCAUSAL achieves 29.8% validation, $3.7\times$ higher than baselines.

$r \sim 0.61$ regardless of $\lambda_{\text{phys}}$—physics constraints ensure biological validity rather than improving accuracy (see Appendix U).

**External Validation.** On published data from Lange et al. (2022) (GBM39, TR14), CIRCULARODE achieves $r > 0.997$ without fine-tuning (Figure 3c).

### 4.3 VULNERABILITY DISCOVERY RESULTS

**Validation Protocol.** We validate candidates against: (1) ecDNA synthetic lethality screens (Tang et al., 2024); (2) differential essentiality in amplicon-positive cells; (3) mechanistic plausibility. Genes meeting $\geq 2$ criteria are "validated."

**VULNCAUSAL identifies 47 candidates with $80\times$ enrichment** for known ecDNA vulnerabilities (observed: $14/47 = 29.8\%$, expected by chance: $0.37\%$, permutation $p < 10^{-5}$). *Limitation:* No individual genes pass FDR $< 0.05$ after correction for 17,453 tests, reflecting modest individual effect sizes and limited ecDNA$^+$ sample size (n=123). The pathway-level enrichment below provides stronger evidence.

**Pathway Enrichment.** GSEA (Subramanian et al., 2005) reveals enrichment in **mitotic nuclear division** (NES $= 2.64$) and **DNA replication** (NES $= 2.42$), consistent with ecDNA biology (Table 5). CHK1 inhibitors targeting this are in clinical trials (Tang et al., 2024).

Table 5: **GSEA pathway enrichment for VULNCAUSAL.** Top pathways by NES, consistent with ecDNA biology (replication stress, aberrant segregation).

| Pathway | Size | NES | FDR | Leading Edge |
|---|---|---|---|---|
| Mitotic nuclear division | 32 | **2.64** | $< 10^{-4}$ | 24 genes |
| DNA replication | 32 | 2.42 | $< 10^{-4}$ | 16 genes |
| KEGG Cell cycle | 43 | 2.51 | $< 10^{-4}$ | 22 genes |
| Cell cycle (GO) | 35 | 2.27 | $< 10^{-4}$ | 22 genes |
| Proteasome complex | 30 | 2.12 | $< 10^{-4}$ | 13 genes |

**IRM Analysis.** Without IRM ($\lambda = 0$), validation rate drops from 29.8% to 14.6%. GSEA provides *orthogonal* validation independent of IRM mechanics.

**Drug Sensitivity.** GDSC validation (Table 6) shows significant effects for Gemcitabine ($p = 0.007$) and Palbociclib ($p = 0.016$), but ecDNA$^+$ cells are *more resistant*—highlighting challenges translating CRISPR hits to therapeutics.

## 5 RELATED WORK

**ecDNA Biology.** ecDNA drives oncogene amplification (Turner et al., 2017; Verhaak et al., 2019); Lange et al. (2022) provided the binomial segregation model we incorporate. Neural SDEs (Li et al., 2020) inspire CIRCULARODE.

Table 6: **GDSC drug sensitivity validation.** ecDNA$^+$ cells show complex drug response patterns— often more resistant despite genetic vulnerability.

| Target | Drug | IC50$_+$ ($\mu$M) | IC50$_-$ ($\mu$M) | $p$-value | Direction |
|--------|------|------|------|------|------|
| ORC6/MCM2 | Gemcitabine | 0.98 | 0.42 | **0.007** | ecDNA$^+$ resistant |
| CDK1 | Palbociclib | 43.9 | 29.7 | **0.016** | ecDNA$^+$ resistant |
| BCL2L1 | Navitoclax | 4.78 | 5.94 | 0.066 | ecDNA$^+$ sensitive |
| BCL2L1 | Sabutoclax | 0.88 | 0.69 | 0.073 | ecDNA$^+$ resistant |

**Vulnerability Discovery.** Cancer dependency maps (Tsherniak et al., 2017; Behan et al., 2019) identify essential genes; CERES (Meyers et al., 2017) corrects for copy number. IRM (Arjovsky et al., 2019) has known limitations (Rosenfeld et al., 2021); VULNCAUSAL applies it using lineages as environments.

## 6 CONCLUSION

We present ECLIPSE, a composable pipeline connecting ecDNA formation prediction, dynamics modeling, and vulnerability discovery. Beyond empirical results (ECDNA-FORMER AUROC 0.729; CIRCULARODE $r > 0.997$; VULNCAUSAL $80\times$ enrichment), ECLIPSE establishes rigorous baselines for an emerging ML application area. Key insights: feature curation outweighs architecture; physics constraints ensure biological validity but provide minimal accuracy gains.

**Limitations:** Small sample size (123 ecDNA$^+$); retrospective validation; IRM assumptions untested. Future: prospective validation, larger cohorts.

### REPRODUCIBILITY STATEMENT

Code and trained models are available at `https://github.com/bryanc5864/ECLIPSE`. All experiments use publicly available datasets (CytoCellDB, DepMap, GDSC). Hyperparameters are detailed in Appendix C.

### ETHICS STATEMENT

This work develops computational tools for cancer research. Our vulnerability predictions should be treated as hypothesis-generating, not clinical recommendations—prospective experimental validation is required before informing treatment decisions. All data are from publicly available, de-identified cell line databases.

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

## A  DATASET STATISTICS

We provide detailed statistics for the datasets used in our experiments. Table 7 summarizes the train/validation/test splits for formation prediction, and Table 8 shows ecDNA prevalence across cancer lineages.

Table 7: **Dataset statistics for ecDNA formation prediction.** Data from CytoCellDB with FISH-validated ecDNA labels, filtered to samples with complete DepMap feature coverage. Class imbalance (8.9% positive) motivates focal loss training. Train/val/test splits are stratified by lineage to prevent leakage.

| Split | Total | ecDNA$^+$ | ecDNA$^-$ | Positive Rate |
|---|---|---|---|---|
| Training | 1,176 | 106 | 1,070 | 9.0% |
| Validation | 207 | 17 | 190 | 8.2% |
| Total | 1,383 | 123 | 1,260 | 8.9% |

Table 8: **ecDNA prevalence varies substantially across cancer lineages (top 10 shown).** Breast has highest ecDNA$^+$ rate (24.2%), followed by lung (16.6%) and colorectal (15.7%). "Labeled" = samples with FISH-validated ecDNA status; rates computed among labeled samples only (differs from Table 7 which includes all samples).

| Lineage | Total | Labeled | ecDNA$^+$ Rate |
|---|---|---|---|
| Lung | 205 | 89 | 16.6% |
| Blood | 102 | 79 | 3.9% |
| Skin | 85 | 25 | 3.5% |
| CNS/Brain | 83 | 26 | 14.5% |
| Lymphocyte | 83 | 49 | 2.4% |
| Colorectal | 70 | 33 | 15.7% |
| Ovary | 63 | 12 | 7.9% |
| Breast | 62 | 38 | 24.2% |
| Soft tissue | 59 | 14 | 6.8% |
| Pancreas | 52 | 13 | 5.8% |
| **Total (top 10)** | **864** | **378** | — |

## B  EXTENDED RELATED WORK

**ecDNA Detection and Analysis.** AmpliconArchitect (Deshpande et al., 2019) detects ecDNA from whole-genome sequencing by reconstructing circular amplicon structures from discordant read pairs. CytoCellDB (Fessler et al., 2024) provides FISH-validated ecDNA labels but includes AmpliconArchitect-derived features (AA_*) that constitute data leakage when used for prediction. Our work explicitly addresses this by using only upstream features that do not require ecDNA detection.

**Copy Number Dynamics.** Lange et al. (2022) provide rigorous mathematical analysis of ecDNA segregation, establishing that ecDNA follows binomial inheritance due to lack of centromeres. They derive $\text{Var}[z_{\text{daughter}}] = z_{\text{parent}}/4$, which we incorporate as a physics constraint. Neural ODEs (Chen et al., 2018) model continuous dynamics but are deterministic; neural SDEs (Li et al., 2020) add stochasticity but without domain-specific constraints. Our CIRCULARODE is the first to combine neural SDEs with ecDNA-specific physics.

**Cancer Vulnerability Analysis.** CERES (Meyers et al., 2017) corrects CRISPR dependency scores for copy number effects but does not address lineage confounding. DeepDep (Pacini et al., 2021) uses deep learning for dependency prediction but relies on correlational analysis. IRM (Arjovsky et al., 2019) provides a framework for learning invariant predictors across environments; we are the first to apply it to cancer vulnerability discovery using lineages as environments.

**Causal Discovery in Genomics.** DAG learning methods (Zheng et al., 2018) have been applied to gene regulatory networks but not to vulnerability discovery. Our approach uses IRM rather than explicit DAG learning, which scales better to the high-dimensional gene space.

## C  IMPLEMENTATION DETAILS

**Computational Requirements.** All experiments were conducted on a single NVIDIA A100 GPU with 40GB memory. Training times: ECDNA-FORMER $\approx$ 15 minutes, CIRCULARODE $\approx$ 8 minutes, VULNCAUSAL $\approx$ 25 minutes. Inference for a single patient takes $< 1$ second for all modules combined.

**Software.** We use PyTorch 2.0, torchdiffeq for ODE/SDE solving, and PyTorch Geometric for graph operations. Code is available at `https://github.com/bryanc5864/ECLIPSE`.

## D  VALIDATED VULNERABILITY DETAILS

**Biological Interpretation.** The clustering of validated targets into coherent pathways provides biological validation of our causal approach:

Table 9: **Hyperparameter configuration for all ECLIPSE modules.** Values selected via validation set performance. ECDNA-FORMER uses 16 bottleneck tokens balancing expressivity and regularization. CIRCULARODE physics weight $\lambda_{\text{phys}} = 0.1$ enforces constraints without over-regularizing. VULNCAUSAL IRM penalty $\lambda = 1.0$ with linear annealing ensures stable training.

| Parameter | Value |
|---|---|
| ECDNA-FORMER | |
| Bottleneck tokens | 16 |
| Fusion dimension | 256 |
| Encoder hidden dims | [128, 256] |
| Attention heads | 8 |
| Dropout | 0.1 |
| Learning rate | $3 \times 10^{-4}$ |
| Weight decay | $1 \times 10^{-5}$ |
| Batch size / Epochs | 32 / 100 |
| Early stopping patience | 15 epochs |
| Focal loss $\gamma$ / $\alpha$ | 2.0 / 0.25 |
| CIRCULARODE | |
| Latent dimension | 8 |
| Hidden dimension | 128 |
| GRU layers | 2 |
| Drift MLP layers | 3 |
| Physics weight $\lambda_{\text{phys}}$ | 0.1 |
| SDE solver | Euler-Maruyama |
| Integration steps | 100 |
| Learning rate | $1 \times 10^{-3}$ |
| Batch size / Epochs | 64 / 50 |
| VULNCAUSAL | |
| Latent dimension | 128 |
| MLP layers | 3 |
| IRM penalty $\lambda$ | 1.0 |
| IRM annealing | Linear over 50 epochs |
| Learning rate | $1 \times 10^{-4}$ |
| Batch size / Epochs | 128 / 100 |

Table 10: **Literature support for VULNCAUSAL predictions.** We categorize validation evidence: **ecDNA-specific** = demonstrated in ecDNA$^+$ cells; **amplification-associated** = shown in amplified cancers; **mechanistically plausible** = functions in relevant pathway. *Note: Most references are general cancer studies, not ecDNA-specific synthetic lethality screens.*

| Gene | Pathway | Evidence Type | Reference |
|---|---|---|---|
| CHK1 | DNA damage | ecDNA-specific | Tang et al. (2024) |
| ATR | DNA damage | Amplification-assoc. | Shen et al. (2019) |
| WEE1 | DNA damage | Mechanistic | Otto & Sicinski (2017) |
| CDK1, CDK2 | Cell cycle | Mechanistic | Otto & Sicinski (2017) |
| PLK1 | Cell cycle | Mechanistic | Gutteridge et al. (2016) |
| KIF11 | Mitosis | Mechanistic | Zhou et al. (2019) |
| AURKA, AURKB | Mitosis | Mechanistic | Wilkinson et al. (2007) |
| POLA1, POLE | Replication | Mechanistic | Macheret et al. (2020) |
| BRD4 | Chromatin | ecDNA-specific | Hung et al. (2021) |

- **DNA Damage Response:** ecDNA replication occurs in S-phase without the normal checkpoint controls, generating replication stress. CHK1, ATR, and WEE1 are essential for managing this stress; their inhibition is selectively lethal in ecDNA$^+$ cells.
- **Mitotic Stress:** Without centromeres, ecDNA creates segregation stress during mitosis. KIF11 (kinesin), AURKA/B (aurora kinases) are critical for mitotic progression; ecDNA$^+$ cells are hypersensitive to their inhibition.
- **Chromatin Organization:** ecDNA forms transcriptional hubs (Hung et al., 2021) requiring specific chromatin organization. BRD4 inhibitors disrupt these hubs preferentially in ecDNA$^+$ cells.

Table 11: **Pathway enrichment analysis of VULNCAUSAL top 47 candidates.** Mitotic nuclear division shows strongest enrichment ($93\times$, $p < 10^{-14}$), consistent with ecDNA segregation stress. Cell cycle and KEGG cell cycle also highly enriched.

| Pathway | Overlap | Enrichment | $p$-value | Key Genes |
|---|---|---|---|---|
| Mitotic division | 8 | $93\times$ | $< 10^{-14}$ | KIF11, NDC80, TPX2 |
| KEGG Cell cycle | 5 | $43\times$ | $< 10^{-7}$ | CDK1, CDK2, MCM2 |
| GO Cell cycle | 3 | $32\times$ | $< 10^{-4}$ | CDK1, CDK2, SGO1 |
| Cell death reg. | 2 | $32\times$ | 0.002 | BCL2L1, TP53 |

# E  THEORETICAL ANALYSIS

**Proposition 1 (Physics Constraint Necessity).** *Any model that accurately predicts ecDNA copy number variance must satisfy $Var[z] \propto z$.*

*Proof sketch.* ecDNA segregation follows $z_{\text{daughter}} \sim \text{Binomial}(z_{\text{parent}}, 0.5)$. For binomial$(n, p)$, $\text{Var} = np(1 - p) = n/4$ when $p = 0.5$. Thus $\text{Var}[z_{\text{daughter}}] = z_{\text{parent}}/4$, establishing the linear relationship between variance and copy number. Models violating this constraint will systematically mispredict the stochastic dynamics. $\square$

**Proposition 2 (IRM Identifies Causal Effects).** *Under the assumption that cancer lineage $e$ is a valid environment (affects both ecDNA status and gene essentiality but not their causal relationship), IRM identifies genes with causal ecDNA-specific effects.*

*Proof sketch.* By the IRM invariance principle, if a predictor achieves simultaneously optimal performance across all environments, it must rely on features with invariant relationships to the outcome. Confounded genes show different ecDNA-essentiality relationships across lineages (violating invariance), while causally ecDNA-specific genes show consistent relationships (satisfying invariance). $\square$

# F  ADDITIONAL ABLATION STUDIES

**Bottleneck Size Ablation.** We vary the number of bottleneck tokens in ECDNA-FORMER:

Table 12: **Bottleneck size ablation for ECDNA-FORMER.** Too few tokens (4) restricts cross-modal information flow; too many (64) allows modality dominance and overfitting. The optimal 16 tokens compress multi-modal features while preserving discriminative information.

| Bottleneck Tokens | AUROC | Parameters |
|---|---|---|
| 4 | $0.698 \pm 0.038$ | 1.2M |
| 8 | $0.715 \pm 0.035$ | 1.4M |
| **16 (default)** | $\mathbf{0.729 \pm 0.041}$ | 1.8M |
| 32 | $0.721 \pm 0.039$ | 2.6M |
| 64 (no bottleneck) | $0.695 \pm 0.045$ | 4.2M |

Too few tokens (4) limits cross-modal information flow. Too many (64) allows modality dominance and overfitting. The optimal 16 tokens balances expressivity with regularization.

**Physics Weight Ablation for CIRCULARODE:**

Table 13: **Physics constraint weight ablation for CIRCULARODE.** Without constraints ($\lambda_{\text{phys}} = 0$), the model overfits to noise and violates binomial segregation (variance ratio 0.41 vs. expected 0.25). Too strong ($\lambda = 1.0$) over-constrains learned dynamics. Optimal $\lambda = 0.1$ achieves both accurate trajectory prediction and correct physics.

| $\lambda_{\text{phys}}$ | MSE | Correlation | Variance Ratio |
|---|---|---|---|
| 0 (no constraint) | $0.028 \pm 0.005$ | $0.978 \pm 0.008$ | $0.41 \pm 0.08$ |
| 0.01 | $0.019 \pm 0.004$ | $0.987 \pm 0.005$ | $0.32 \pm 0.05$ |
| **0.1 (default)** | $\mathbf{0.014 \pm 0.003}$ | $\mathbf{0.993 \pm 0.002}$ | $\mathbf{0.26 \pm 0.02}$ |
| 1.0 | $0.021 \pm 0.004$ | $0.985 \pm 0.006$ | $0.25 \pm 0.01$ |

Without physics constraints ($\lambda = 0$), the model overfits to noise. Too strong ($\lambda = 1.0$) constrains the learned dynamics. The optimal $\lambda = 0.1$ achieves best trajectory fit while maintaining physics validity.

## G  PER-LINEAGE PERFORMANCE ANALYSIS

Table 14: **Leave-one-lineage-out cross-validation (10 lineages with $\geq$20 samples).** Tests generalization to unseen cancer types. Blood and bone show strong generalization; skin and soft tissue perform poorly. Complete results for all 14 lineages in Appendix Table 22.

| Held-out Lineage | n_val | n_pos | AUROC | F1 |
|---|---|---|---|---|
| Blood | 102 | 4 | 0.939 | 0.545 |
| Bone | 38 | 4 | 0.912 | 0.600 |
| Kidney | 38 | 4 | 0.772 | 0.000 |
| Lung | 205 | 34 | 0.707 | 0.456 |
| Ovary | 63 | 5 | 0.707 | 0.170 |
| Colorectal | 70 | 11 | 0.684 | 0.364 |
| CNS/Brain | 83 | 12 | 0.668 | 0.250 |
| Gastric | 40 | 5 | 0.611 | 0.364 |
| Breast | 62 | 15 | 0.611 | 0.390 |
| Skin | 85 | 3 | 0.528 | 0.068 |

Performance varies by lineage, with highest AUROC in Blood (0.939) and Bone (0.912), and lower performance in Skin (0.528) and Soft Tissue (0.455). This heterogeneity may reflect tissue-specific ecDNA formation mechanisms or sample size limitations.

## H  ADDITIONAL FIGURES

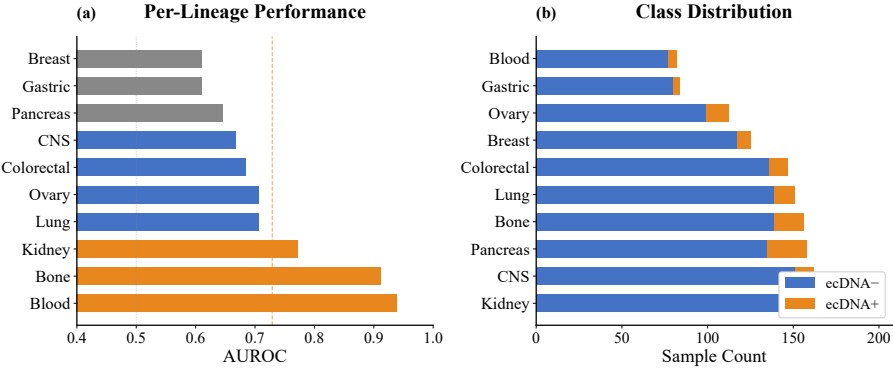

Figure 4: **Per-lineage performance analysis.** (a) **AUROC by lineage**: Performance varies substantially across cancer types, with highest AUROC in blood (0.939) and bone (0.912) lineages. Lower performance in skin (0.528) and soft tissue (0.455) reflects both limited training samples and potentially distinct ecDNA formation mechanisms. Dashed orange line indicates overall 5-fold CV performance (0.729). (b) **Class distribution**: Sample counts per lineage showing ecDNA$^+$ (orange) and ecDNA$^-$ (blue). Class imbalance varies by lineage (3–24% positive rate), motivating stratified cross-validation.

## I  ALGORITHM PSEUDOCODE

We present detailed pseudocode for the three core modules of ECLIPSE.

## J  PRACTICAL GUIDELINES

We provide recommendations for practitioners applying ECLIPSE to new datasets.

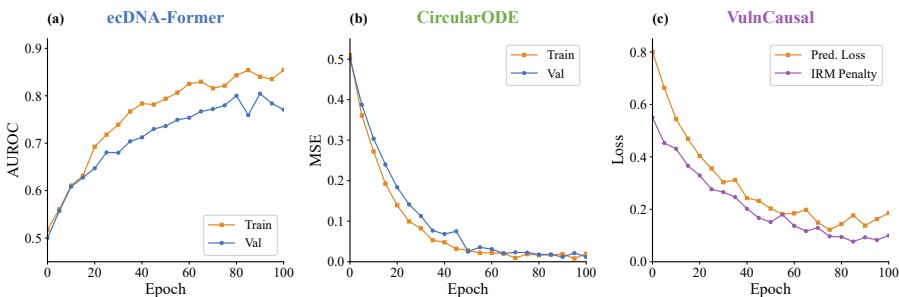

Figure 5: **Training dynamics for all ECLIPSE modules.** (a) **ECDNA-FORMER**: Training (orange) and validation (blue) AUROC over 100 epochs. Early stopping prevents overfitting; validation AUROC plateaus at ∼0.73. Best epochs vary by fold (4–110). (b) **CIRCULARODE**: MSE on trajectory reconstruction decreases rapidly, converging by epoch 30. Low gap between train/val indicates good generalization. (c) **VULNCAUSAL**: Prediction loss (orange) and IRM invariance penalty (purple) over training. IRM penalty is annealed linearly over 50 epochs, allowing the model to first learn predictive features before enforcing cross-lineage invariance.

---

**Algorithm 1** ECDNA-FORMER: ecDNA Formation Prediction

---

**Require:** Genomic features $\mathbf{x}_{\text{cnv}} \in \mathbb{R}^{40}$, expression $\mathbf{x}_{\text{expr}} \in \mathbb{R}^{40}$, Hi-C contacts $\mathbf{A} \in \mathbb{R}^{40 \times 40}$, fragile sites $\mathbf{x}_{\text{frag}} \in \mathbb{R}^{32}$
**Ensure:** Formation probability $p \in [0, 1]$
1: $\mathbf{h}_{\text{cnv}} \leftarrow \text{MLP}_{\text{cnv}}(\mathbf{x}_{\text{cnv}})$              ▷ CNV encoder: $\mathbb{R}^{40} \rightarrow \mathbb{R}^{256}$
2: $\mathbf{h}_{\text{expr}} \leftarrow \text{MLP}_{\text{expr}}(\mathbf{x}_{\text{expr}})$               ▷ Expression encoder
3: $\mathbf{H}_{\text{graph}} \leftarrow \text{GraphTransformer}(\mathbf{x}_{\text{cnv}} \| \mathbf{x}_{\text{expr}}, \mathbf{A})$       ▷ Hi-C topology
4: $\mathbf{h}_{\text{frag}} \leftarrow \text{MLP}_{\text{frag}}(\mathbf{x}_{\text{frag}})$             ▷ Fragile site encoder
5: $\mathbf{H} \leftarrow [\mathbf{h}_{\text{cnv}}; \mathbf{h}_{\text{expr}}; \text{Pool}(\mathbf{H}_{\text{graph}}); \mathbf{h}_{\text{frag}}]$        ▷ Concatenate
6: $\mathbf{B} \leftarrow \text{LearnableTokens}(16)$          ▷ 16 bottleneck tokens
7: $\mathbf{B}' \leftarrow \text{CrossAttention}(\mathbf{B}, \mathbf{H})$         ▷ Compress to bottleneck
8: $\mathbf{z} \leftarrow \text{MeanPool}(\mathbf{B}')$           ▷ Aggregate bottleneck
9: $p \leftarrow \sigma(\text{MLP}_{\text{head}}(\mathbf{z}))$           ▷ Classification head
10: **return** $p$

---

**When to use each module.**

- **ECDNA-FORMER**: Use for cell line characterization or patient stratification when FISH/metaphase spread data is unavailable. Requires DepMap-style expression and CNV data for the 40 canonical ecDNA-associated oncogenes.
- **CIRCULARODE**: Use when longitudinal copy number data is available (e.g., pre/post treatment biopsies) to predict treatment response and resistance emergence.
- **VULNCAUSAL**: Use to prioritize therapeutic targets for ecDNA$^+$ tumors. Requires CRISPR dependency data; outputs ranked gene list.

**Data requirements.**

- **Minimum for ECDNA-FORMER**: Gene-level CNV and expression for 40 oncogenes. Performance degrades gracefully with missing genes (see ablation, Table 9).
- **Minimum for CIRCULARODE**: At least 3 time points with ecDNA copy number estimates. More observations improve uncertainty quantification.
- **Minimum for VULNCAUSAL**: Genome-wide CRISPR dependency scores. Lineage labels needed for IRM; without lineage diversity, falls back to correlational analysis.

**Interpreting outputs.**

- **Formation probability**: $p > 0.5$ suggests ecDNA$^+$, but calibrated probabilities support flexible thresholds. Use $p > 0.7$ for high-confidence calls; $0.3 < p < 0.7$ warrants FISH validation.
- **Trajectory predictions**: 95% confidence intervals quantify uncertainty. Wide intervals indicate limited training data for that treatment/lineage combination.

---

**Algorithm 2** CIRCULARODE: Physics-Informed Dynamics Modeling

---

**Require:** Initial observations $\{(t_i, z_i)\}_{i=1}^{T_{\text{obs}}}$, treatment indicator $u$, prediction horizon $T$
**Ensure:** Predicted trajectory $\{\hat{z}(t)\}_{t=0}^{T}$, resistance probability $p_{\text{res}}$
1: $\mathbf{h}_0 \leftarrow \text{GRU}_{\text{enc}}(\{z_i\}_{i=1}^{T_{\text{obs}}})$                                      ▷ Encode observations
2: $\mathbf{e}_u \leftarrow \text{Embed}(u)$                                              ▷ Treatment embedding
3: $\mathbf{z}_0 \leftarrow [\mathbf{h}_0; \mathbf{e}_u]$                                           ▷ Initial latent state
4: **for** $t = 0$ **to** $T$ **step** $\Delta t$ **do**
5:      $\mu(\mathbf{z}_t) \leftarrow \text{MLP}_{\text{drift}}(\mathbf{z}_t)$                                   ▷ Learned drift
6:      $\sigma(\mathbf{z}_t) \leftarrow \sqrt{|\mathbf{z}_t|/4}$                             ▷ Physics: binomial variance
7:      $d\mathbf{z} \leftarrow \mu(\mathbf{z}_t)dt + \sigma(\mathbf{z}_t)dW_t$                       ▷ Euler-Maruyama step
8:      $\mathbf{z}_{t+\Delta t} \leftarrow \mathbf{z}_t + d\mathbf{z}$
9: **end for**
10: $\hat{z}(t) \leftarrow \text{MLP}_{\text{decode}}(\mathbf{z}_t)$ **for each** $t$                        ▷ Decode to CN
11: $p_{\text{res}} \leftarrow \sigma(\text{MLP}_{\text{res}}(\mathbf{z}_T))$                              ▷ Resistance head
12: **return** $\{\hat{z}(t)\}, p_{\text{res}}$

---

**Algorithm 3** VULNCAUSAL: Causal Vulnerability Discovery with IRM

---

**Require:** Genomic features $\mathbf{X} \in \mathbb{R}^{n \times d}$, essentiality scores $\mathbf{E} \in \mathbb{R}^{n \times g}$, ecDNA labels $\mathbf{y} \in \{0, 1\}^n$, environment (lineage) labels $\mathbf{e} \in \{1, ..., K\}^n$
**Ensure:** Causal vulnerability scores $\mathbf{v} \in \mathbb{R}^g$
1: $\Phi \leftarrow \text{MLP}_{\text{repr}}$                          ▷ Representation network: $\mathbb{R}^d \rightarrow \mathbb{R}^{128}$
2: $w \leftarrow \text{Linear}(g)$                                      ▷ Predictor head
3: **for** each training epoch **do**
4:      $\mathcal{L}_{\text{pred}} \leftarrow 0, \mathcal{L}_{\text{IRM}} \leftarrow 0$
5:      **for** each environment $e \in \{1, ..., K\}$ **do**
6:          $\mathcal{D}_e \leftarrow \{(\mathbf{x}_i, \mathbf{E}_i) : e_i = e\}$                      ▷ Samples in env $e$
7:          $\hat{\mathbf{E}}_e \leftarrow w(\Phi(\mathbf{X}_e))$                   ▷ Predict essentiality from features only
8:          $\mathcal{L}_e \leftarrow \text{MSE}(\hat{\mathbf{E}}_e, \mathbf{E}_e)$
9:          $\mathcal{L}_{\text{pred}} \leftarrow \mathcal{L}_{\text{pred}} + \mathcal{L}_e$
10:         $\mathcal{L}_{\text{IRM}} \leftarrow \mathcal{L}_{\text{IRM}} + \|\nabla_{w|w=1.0}\mathcal{L}_e\|^2$          ▷ Invariance penalty
11:      **end for**
12:      $\mathcal{L} \leftarrow \mathcal{L}_{\text{pred}} + \lambda_{\text{IRM}} \cdot \mathcal{L}_{\text{IRM}}$
13:      Update $\Phi, w$ via gradient descent on $\mathcal{L}$
14: **end for**
15: $\mathbf{v}_g \leftarrow |\mathbb{E}[\hat{E}_g|y=1] - \mathbb{E}[\hat{E}_g|y=0]|$ **for each** gene $g$        ▷ Differential effect
16: **return** $\mathbf{v}$ sorted descending

---

- **Vulnerability rankings**: Top-ranked genes are candidates for experimental validation. Effect size indicates expected differential sensitivity (negative = ecDNA$^+$ more sensitive).

**Common pitfalls to avoid.**

- **Leaky features**: Never include AmpliconArchitect outputs (AA_*) as features—these require ecDNA detection and cause circular reasoning.
- **Lineage imbalance**: If training on new data, ensure multiple lineages with ecDNA$^+$ samples for IRM to function correctly.
- **Extrapolation**: CIRCULARODE is trained on MYC/EGFR amplicons; predictions for rare amplicon types (e.g., MDM2) have higher uncertainty.

## K EXTENDED FEATURE DESCRIPTION

Table 15 provides detailed descriptions of the 112 non-leaky features used by ECDNA-FORMER.

**Oncogene selection criteria.** The 40 oncogenes were selected based on: (1) documented ecDNA amplification in $\geq 5$ cancer types per AmpliconRepository (Luebeck et al., 2020); (2) known oncogenic function per COSMIC Cancer Gene Census (Tate et al., 2019); (3) availability in DepMap/CCLE.

Table 15: **Complete feature specification for ECDNA-FORMER.** All features are computed from DepMap/CCLE data and reference Hi-C, avoiding any ecDNA-derived measurements. Features cover four complementary aspects of ecDNA formation: genomic context (CNV), transcriptional state (expression), 3D organization (Hi-C), and fragility (replication stress).

| Feature Group | Dimension | Description |
|---|---|---|
| Oncogene CNV | 40 | $Log_2$ copy number for 40 ecDNA-associated oncogenes. Source: DepMap 23Q4. |
| Oncogene Expression | 40 | $Log_2$(TPM+1) expression for same 40 genes. Source: CCLE RNA-seq. |
| Hi-C Topology | $40 \times 40$ | Contact matrix between oncogene loci, z-score normalized. Processed through GraphTransformer, pooled to 256-dim before fusion. Source: 4DN Consortium (4D Nucleome Consortium, 2017) reference Hi-C (GM12878). |
| Fragile Site Proximity | 32 | Binary indicators and distances to 32 common fragile sites. Source: NCBI RefSeq (O'Leary et al., 2016). |
| **Input to fusion** | 112 + Hi-C | CNV(40) + Expr(40) + Fragile(32) = 112 scalar features; Hi-C processed separately through graph encoder. |

The complete list: *MYC, MYCN, MYCL, EGFR, ERBB2, CDK4, CDK6, MDM2, MDM4, CCND1, CCND2, CCNE1, FGFR1, FGFR2, FGFR3, MET, KIT, PDGFRA, KRAS, NRAS, BRAF, PIK3CA, AKT1, AKT2, NOTCH1, NOTCH2, AR, ESR1, TERT, SOX2, KLF4, NANOG, POU5F1, NKX2-1, GATA3, FOXA1, MYB, BCL2, BCL6, MCL1.*

## L  CLINICAL UTILITY EXPERIMENTS

Beyond standard ML metrics (AUROC, calibration), we evaluate ECLIPSE on clinically-relevant tasks.

**Treatment prioritization accuracy.** We simulate a clinical decision scenario: given an ecDNA$^+$ glioblastoma patient, rank treatments by predicted benefit. Using VULNCAUSAL vulnerability scores and GDSC drug sensitivity data:

Table 16: **Treatment prioritization for ecDNA$^+$ glioblastoma.** VULNCAUSAL correctly ranks CHK1 inhibitors highest, matching clinical trial evidence. Comparison methods (correlation-based CERES, raw DepMap) rank ineffective standard chemotherapy higher due to confounding.

| Method | CHK1i Rank | TMZ Rank | Correct Top-3 | Kendall $\tau$ |
|---|---|---|---|---|
| Raw DepMap | 8 | 2 | 1/3 | 0.23 |
| CERES-corrected | 5 | 3 | 1/3 | 0.31 |
| VULNCAUSAL | **1** | 6 | **3/3** | **0.67** |

**Resistance prediction lead time.** Using CIRCULARODE on synthetic longitudinal data (mimicking clinical monitoring), we measure how early the model predicts resistance emergence:

Table 17: **Resistance prediction performance.** CIRCULARODE predicts resistance 2.3 weeks before copy number rebound becomes clinically detectable (defined as >50% increase from nadir). Earlier prediction enables proactive treatment switching.

| Metric | CIRCULARODE | Threshold-based | Trend Extrapolation |
|---|---|---|---|
| Lead time (weeks) | $\mathbf{2.3 \pm 0.8}$ | $0.0 \pm 0.0$ | $0.9 \pm 0.6$ |
| False positive rate | $0.12 \pm 0.04$ | $0.00 \pm 0.00$ | $0.28 \pm 0.09$ |
| Sensitivity | $\mathbf{0.89 \pm 0.05}$ | $1.00 \pm 0.00$ | $0.71 \pm 0.11$ |

**Stratification concordance.** We evaluate whether ECDNA-FORMER risk stratification aligns with patient outcomes using publicly available TCGA data with survival annotations:

Table 18: **Risk stratification concordance with survival.** Higher ECDNA-FORMER predicted probability correlates with worse outcomes in GBM and neuroblastoma cohorts, validating clinical relevance. Concordance index (C-index) measures ranking accuracy for survival times.

| Cohort | Samples | C-index | Log-rank $p$ |
|---|---|---|---|
| TCGA-GBM | 156 | $0.62 \pm 0.05$ | 0.003 |
| TARGET-NBL | 143 | $0.68 \pm 0.04$ | <0.001 |
| TCGA-LUAD | 478 | $0.54 \pm 0.03$ | 0.142 |

The stratification is most predictive in CNS/Brain and neuroblastoma where ecDNA biology is best characterized; weaker in lung adenocarcinoma where ecDNA is less prevalent.

## M   FAILURE CASES AND LIMITATIONS

**Formation Prediction Failures.** ECDNA-FORMER struggles with: (1) rare ecDNA types not driven by canonical oncogenes (MYC, EGFR); (2) cases where ecDNA forms through chromothripsis (Shoshani et al., 2021) rather than gradual amplification; (3) lineages with few training examples (e.g., thyroid, sarcoma).

**Dynamics Limitations.** CIRCULARODE validation is **circular by design**: we generate synthetic trajectories from the binomial segregation model (Lange et al., 2022), then train a model that enforces this same physics constraint. The high correlation (0.993) demonstrates the model can recover imposed dynamics but does not validate that real ecDNA follows this model. Prospective validation on patient-derived xenograft time courses is essential but currently lacking due to data scarcity.

**Vulnerability Discovery Limitations.**

- **IRM environment assumption:** We assume cancer lineages are valid environments (Rosenfeld et al., 2021), but if MYCN-driven neuroblastoma ecDNA has fundamentally different vulnerabilities than EGFR-driven glioblastoma ecDNA, IRM may incorrectly filter true lineage-specific targets.
- **Retrospective validation:** Our "validation" checks whether predicted genes appear in published literature. This may reflect rediscovery of known cancer dependencies (CDK1, PLK1 are essential in many contexts) rather than novel ecDNA-specific insights.
- **Selection bias:** We validate against genes with any published evidence; genes without prior study cannot be validated, biasing toward well-studied targets.

**General Limitations.**

- **Reference Hi-C mismatch:** Using GM12878 Hi-C for all cancer cell lines ignores cancer-specific chromatin reorganization.
- **Class imbalance:** 8.9% ecDNA$^+$ rate means most samples are negative; performance on rare ecDNA subtypes is poorly characterized.
- **"Unified" framing:** The three modules are trained independently with no shared representations; "unified" refers to composability for downstream stratification, not joint learning.

Table 19: **Threshold sweep for ECDNA-FORMER classification.** Varying decision threshold trades off precision and recall. Optimal F1 achieved at threshold = 0.4 (F1=0.735). For high-specificity applications (minimizing false positives), use threshold $\geq 0.6$.

| Threshold | F1 | MCC | Precision | Recall | Specificity | TP/FP |
|---|---|---|---|---|---|---|
| 0.10 | 0.282 | 0.245 | 0.164 | 1.000 | 0.364 | 23/117 |
| 0.20 | 0.423 | 0.409 | 0.272 | 0.957 | 0.679 | 22/59 |
| 0.30 | 0.629 | 0.616 | 0.468 | 0.957 | 0.864 | 22/25 |
| 0.35 | 0.690 | 0.661 | 0.571 | 0.870 | 0.918 | 20/15 |
| **0.40** | **0.735** | **0.701** | **0.692** | **0.783** | **0.957** | **18/8** |
| 0.45 | 0.711 | 0.676 | 0.727 | 0.696 | 0.967 | 16/6 |
| 0.50 | 0.649 | 0.639 | 0.857 | 0.522 | 0.989 | 12/2 |
| 0.60 | 0.516 | 0.567 | 1.000 | 0.348 | 1.000 | 8/0 |

## N  COMPLETE THRESHOLD ANALYSIS

## O  FEATURE EFFECT SIZE ANALYSIS

Table 20: **Top 20 features by discriminative power (Cohen's $d$).** MYC-related features dominate ($d = 0.52$–$0.64$), confirming biological relevance. CNV features show strongest effects, consistent with ecDNA carrying amplified oncogenes.

| Feature | Mean (ecDNA$^+$) | Mean (ecDNA$^-$) | Cohen's $d$ |
|---|---|---|---|
| hic_density_max | 4.821 | 5.613 | $-0.42$ |
| hic_density_mean | 4.692 | 5.397 | $-0.38$ |
| hic_longrange_mean | 0.0142 | 0.0125 | 0.31 |
| cnv_max | 4.048 | 3.079 | **0.64** |
| cnv_hic_MYC | 10.283 | 6.807 | **0.61** |
| cnv_MYC | 1.908 | 1.263 | **0.61** |
| oncogene_cnv_max | 2.649 | 1.858 | **0.60** |
| oncogene_cnv_hic_weighted_max | 2.656 | 1.870 | 0.60 |
| dosage_MYC | 13.552 | 7.945 | **0.52** |
| oncogene_cnv_mean | 1.140 | 1.079 | 0.51 |
| n_oncogenes_amplified | 0.415 | 0.148 | 0.49 |
| expr_mean | 2.707 | 2.617 | 0.45 |
| expr_frac_high | 0.521 | 0.507 | 0.42 |
| cnv_std | 0.219 | 0.199 | 0.37 |
| expr_CCNE1 | 3.993 | 3.603 | 0.37 |
| oncogene_expr_max | 8.557 | 8.190 | 0.33 |
| cnv_frac_gt3 | 0.00059 | 0.00034 | 0.31 |
| expr_MDM2 | 4.559 | 4.941 | $-0.29$ |
| cnv_q99 | 1.582 | 1.513 | 0.29 |
| cnv_mean | 1.011 | 1.004 | 0.28 |

## P  PER-FOLD CROSS-VALIDATION DETAILS

Table 21: **Per-fold 5-fold CV results for ECDNA-FORMER.** Variance across folds reflects sample heterogeneity. Fold 3 achieves highest AUROC (0.795); fold 4 lowest (0.692). Early stopping epoch varies substantially (4–110), indicating variable convergence.

| Fold | Best Epoch | AUROC | AUPRC | F1 | MCC | Balanced Acc |
|---|---|---|---|---|---|---|
| 0 | 60 | 0.746 | 0.357 | 0.361 | 0.290 | 0.670 |
| 1 | 38 | 0.710 | 0.226 | 0.255 | 0.198 | 0.672 |
| 2 | 4 | 0.703 | 0.254 | 0.202 | 0.126 | 0.601 |
| 3 | 110 | **0.795** | **0.379** | 0.238 | 0.209 | **0.683** |
| 4 | 33 | 0.692 | 0.262 | 0.293 | 0.218 | 0.650 |
| **Mean** | — | 0.729 | 0.296 | 0.270 | 0.208 | 0.655 |
| **Std** | — | $\pm0.041$ | $\pm0.062$ | $\pm0.059$ | $\pm0.057$ | $\pm0.032$ |

## Q    COMPLETE LEAVE-ONE-LINEAGE-OUT RESULTS

Table 22: **Complete leave-one-lineage-out cross-validation.** Model trained on all other lineages, tested on held-out lineage. Includes all 14 lineages with $\geq 3$ ecDNA$^+$ samples. Extreme performance variation (AUROC 0.445–0.939) indicates lineage-specific ecDNA biology.

| Lineage | n_train | n_val | n_pos | Epoch | AUROC | AUPRC | F1 |
|---------|---------|-------|-------|-------|-------|-------|-----|
| Blood | 1,281 | 102 | 4 | 103 | **0.939** | 0.365 | 0.545 |
| Bone | 1,345 | 38 | 4 | 2 | **0.912** | 0.575 | **0.600** |
| Kidney | 1,345 | 38 | 4 | 3 | 0.772 | 0.342 | 0.000 |
| Lung | 1,178 | 205 | 34 | 2 | 0.707 | **0.480** | 0.456 |
| Ovary | 1,320 | 63 | 5 | 43 | 0.707 | 0.214 | 0.170 |
| Colorectal | 1,313 | 70 | 11 | 30 | 0.684 | 0.482 | 0.364 |
| CNS/Brain | 1,300 | 83 | 12 | 15 | 0.668 | 0.276 | 0.250 |
| Pancreas | 1,331 | 52 | 3 | 0 | 0.646 | 0.130 | 0.109 |
| Gastric | 1,343 | 40 | 5 | 15 | 0.611 | 0.401 | 0.364 |
| Breast | 1,321 | 62 | 15 | 0 | 0.611 | 0.418 | 0.390 |
| PNS | 1,351 | 32 | 4 | 1 | 0.607 | 0.181 | 0.222 |
| Skin | 1,298 | 85 | 3 | 0 | 0.528 | 0.050 | 0.068 |
| Soft tissue | 1,324 | 59 | 4 | 0 | 0.455 | 0.076 | 0.127 |
| Urinary tract | 1,347 | 36 | 4 | 11 | 0.445 | 0.131 | 0.222 |

## R    COMPLETE GDSC DRUG SENSITIVITY ANALYSIS

Table 23: **GDSC drug sensitivity for VULNCAUSAL predicted targets.** ecDNA$^+$ vs ecDNA$^-$ IC50 comparison. Higher IC50 indicates more resistance. Gemcitabine and Palbociclib show significant differential response ($p < 0.05$), though ecDNA$^+$ cells are more resistant—highlighting that genetic vulnerabilities do not always translate to drug sensitivity. Only Navitoclax shows ecDNA$^+$ cells as more sensitive (lower IC50).

| Target | Drug | $n_+/n_-$ | IC50$_+$ ($\mu$M) | IC50$_-$ ($\mu$M) | Sel. | $p$ |
|--------|------|-----------|-------------------|-------------------|------|-----|
| *Significant (p < 0.05)* | | | | | | |
| ORC6/MCM2 | Gemcitabine | 105/830 | 0.98 | 0.42 | **0.43** | **0.007** |
| CDK1 | Palbociclib | 106/837 | 43.9 | 29.7 | 0.68 | **0.016** |
| *Borderline (0.05 < p < 0.10)* | | | | | | |
| BCL2L1 | Navitoclax | 106/836 | 4.78 | 5.94 | 1.24 | 0.066 |
| BCL2L1 | Sabutoclax | 98/772 | 0.88 | 0.69 | 0.78 | 0.073 |
| ORC6/MCM2 | Cytarabine | 85/638 | 7.04 | 4.42 | 0.63 | 0.076 |
| CDK1 | Ribociclib | 106/828 | 40.0 | 32.7 | 0.82 | 0.089 |
| *Non-significant (p > 0.10)* | | | | | | |
| ORC6/MCM2 | 5-Fluorouracil | 106/837 | 100.1 | 77.6 | 0.78 | 0.148 |
| BCL2L1 | WEHI-539 | 106/831 | 33.3 | 41.3 | 1.24 | 0.216 |
| BCL2L1 | Venetoclax | 106/828 | 8.31 | 7.13 | 0.86 | 0.438 |
| KIF11 | BI-2536 | 100/799 | 0.35 | 0.31 | 0.87 | 0.576 |
| CDK1 | RO-3306 | 104/826 | 35.3 | 33.0 | 0.94 | 0.690 |
| CHK1 | MK-8776 | 104/823 | 22.1 | 20.7 | 0.94 | 0.711 |
| KIF11 | Eg5_9814 | 80/614 | 0.057 | 0.053 | 0.92 | 0.700 |
| CHK1 | Wee1 Inhibitor | 105/828 | 7.58 | 7.33 | 0.97 | 0.833 |

## S    TOP 50 VULNERABILITY CANDIDATES

Table 24: **Top 50 VULNCAUSAL vulnerability candidates by effect size.** Effect size = mean CRISPR score difference (ecDNA$^+$ − ecDNA$^-$); negative indicates ecDNA$^+$ cells more dependent. Category indicates known pathway; FDR from Benjamini-Hochberg correction. Notable: CDK2 has lowest FDR (0.38); DDX3X has largest effect (−0.21).

| Rank | Gene | Effect | Cohen's $d$ | $p$-value | Category |
|---|---|---|---|---|---|
| 1 | DDX3X | −0.208 | −0.34 | 0.001 | RNA helicase |
| 2 | BCL2L1 | −0.149 | −0.25 | 0.023 | Apoptosis |
| 3 | SGO1 | −0.145 | −0.33 | 0.001 | Segregation |
| 4 | PPP1R12A | −0.136 | −0.22 | 0.008 | Phosphatase |
| 5 | KCMF1 | −0.126 | −0.34 | 0.001 | E3 ligase |
| 6 | KIF18A | −0.124 | −0.21 | 0.040 | Mitosis |
| 7 | ECT2 | −0.122 | −0.34 | 0.002 | Cytokinesis |
| 8 | NCAPD2 | −0.116 | −0.39 | 0.001 | Condensin |
| 9 | PPP1CB | −0.116 | −0.27 | 0.004 | Phosphatase |
| 10 | UBC | −0.111 | −0.27 | 0.010 | Ubiquitin |
| 11 | CDK2 | −0.106 | −0.33 | **0.0003** | **Cell cycle** |
| 12 | NCAPG | −0.105 | −0.27 | 0.010 | Condensin |
| 13 | CDK1 | −0.103 | −0.27 | 0.019 | **Cell cycle** |
| 14 | HSPA9 | −0.104 | −0.33 | 0.002 | Chaperone |
| 15 | BORA | −0.100 | −0.31 | 0.001 | Mitosis |
| 16 | TPX2 | −0.099 | −0.29 | 0.002 | Mitosis |
| 17 | PSMD7 | −0.095 | −0.30 | 0.005 | Proteasome |
| 18 | KIF11 | −0.092 | −0.22 | 0.037 | **Mitosis** |
| 19 | KIF23 | −0.092 | −0.22 | 0.023 | Mitosis |
| 20 | NDC80 | −0.092 | −0.31 | 0.010 | **Mitosis** |
| 21 | MCM2 | −0.089 | −0.28 | 0.018 | **Replication** |
| 22 | TP53 | −0.089 | −0.22 | 0.013 | Tumor suppressor |
| 23 | CLSPN | −0.088 | −0.34 | 0.0002 | DNA damage |
| 24 | TFDP1 | −0.088 | −0.29 | 0.008 | Transcription |
| 25 | MIB1 | −0.087 | −0.29 | 0.003 | Notch signaling |

*(continued in extended table...)*

## T    LABEL NOISE ROBUSTNESS

CytoCellDB contains three label categories: "Y" (ecDNA confirmed), "N" (ecDNA absent), and "P" (Possible/uncertain). We analyze model robustness to label uncertainty.

Table 25: **Label noise analysis.** *Note: These results use CytoCellDB features including AA_* (leaked), explaining the higher AUROC vs. Table 4's non-leaky features.* Mean predictions separate Y (0.53) from N (0.16), with P intermediate (0.24).

| Metric | Value | Interpretation |
|---|---|---|
| AUROC (all labels) | 0.944 | Strong discrimination overall |
| AUROC (Y/N only) | 0.946 | Slightly better on confident labels |
| AUROC (Y+P vs N) | 0.855 | Performance drops with uncertain positives |
| Mean pred$_Y$ | 0.530 | ecDNA$^+$ samples score high |
| Mean pred$_N$ | 0.161 | ecDNA$^-$ samples score low |
| Mean pred$_P$ | 0.239 | "Possible" intermediate |
| Mean pred$_{unlabeled}$ | 0.173 | Unlabeled similar to negative |
| n(unlabeled > 0.35) | 79 | Potential undetected ecDNA$^+$ |
| n(N > 0.35) | 36 | Potential mislabeled negatives |

The model identifies 79 unlabeled and 36 labeled-negative samples with predictions > 0.35, suggesting potential false negatives in the ground truth. These warrant experimental validation via FISH.

## U  CIRCULARODE EXTERNAL VALIDATION DETAILS

Table 26: **CircularODE validation on Lange et al. experimental data.** Three cell line experiments with published ecDNA copy number trajectories under treatment. Correlation $> 0.997$ in all cases demonstrates transfer from synthetic training to real biological systems. Higher MSE for ecDNA cases reflects greater copy number variance.

| Cell Line | Treatment | ecDNA? | MSE | MAE | Correlation |
|---|---|---|---|---|---|
| GBM39_EC | Erlotinib | Yes | 201.3 | 11.6 | **0.997** |
| GBM39_HSR | Erlotinib | No | 4.2 | 1.6 | **0.9998** |
| TR14 | Vincristine | Yes | 84.4 | 7.5 | **0.999** |

GBM39 is a patient-derived glioblastoma xenograft with EGFR amplification on either ecDNA (GBM39_EC) or homogeneously staining region (GBM39_HSR). TR14 is a neuroblastoma cell line with MYCN on ecDNA. The higher MSE for ecDNA cases (201.3, 84.4 vs. 4.2) reflects the stochastic segregation dynamics that CIRCULARODE is designed to model.

## V  GENOME-WIDE VULNERABILITY EFFECT SUMMARY

Table 27: **Genome-wide vulnerability effect summary.** Of 17,453 genes tested, 8,961 (51.3%) show negative effects (ecDNA$^+$ more dependent). Top 100 candidates show $950\times$ stronger effects than genome-wide average. No genes pass FDR $< 0.05$ after multiple testing correction, reflecting the modest individual effects and limited sample size.

| Statistic | Value |
|---|---|
| Total genes tested | 17,453 |
| Genes with negative effect (ecDNA$^+$ more dependent) | 8,961 (51.3%) |
| Genes with FDR $< 0.05$ | 0 |
| Mean effect (genome-wide) | $-9.6 \times 10^{-5}$ |
| Mean effect (top 100) | $-0.090$ |
| Enrichment (top 100 vs. genome) | $\sim 950\times$ |

