# OpenReview forum: "ECLIPSE: A Composable Pipeline for Predicting ecDNA Formation, Evolution, and Therapeutic Vulnerabilities in Cancer"
_ICLR.cc/2026/Workshop/FM4Science — ICLR 2026 Workshop FM4Science Poster_

### Official Review · Reviewer_j9tF · 2026-02-14
**Rigorous Baselines and Methodological Corrections for Computational ecDNA Analysis**

**Rating:** 7
**Confidence:** 3

**Review:**

Summary:
This paper presents ECLIPSE, a framework designed to address three fundamental flaws in current computational ecDNA research: circular reasoning (data leakage) in prediction, physics mismatch in dynamics modeling, and lineage confounding in vulnerability discovery.

Strengths:
1. The paper performs a valuable service to the field by explicitly identifying and quantifying the "data leakage" issue in existing benchmarks.

2. The integration of biological priors into the machine learning architecture is well-motivated. Specifically, enforcing the "binomial segregation" variance constraint ($Var = z/4$) in CIRCULARODE  is a smart way to ensure physical plausibility in generative modeling.


3. The paper is commendable for its detailed reporting of negative results (e.g., failure cases in formation prediction ) and resource requirements, which enhances reproducibility.

Weaknesses:


1. While the authors do provide a generalization analysis (Figure 4 ), the results indicate that the model is not yet robust for pan-cancer application. Performance drops significantly in certain lineages (e.g., Skin AUROC 0.528, Soft Tissue AUROC 0.455 ) compared to Blood or Bone. This suggests the current feature set may not capture tissue-specific ecDNA formation mechanisms effectively.


1. The "composable" aspect of the pipeline (Section 3.4) remains largely theoretical. The authors acknowledge that the combined risk scoring equation is "proposed but not experimentally validated". Without demonstrating that the combined pipeline outperforms individual modules for patient stratification, the "unified framework" claim is slightly premature.


1. The label noise analysis (Appendix T) suggests potential false negatives in the ground truth (79 unlabeled samples predicted as high-confidence positives ). While this hints at discovery potential, it also complicates the evaluation of the ECDNA-FORMER module, as the "ground truth" (FISH) itself has limitations that might penalize the model unfairly or obscure true performance.

---

### Official Review · Reviewer_ZRXg · 2026-02-22
**Interesting Work with Strong Potential**

**Rating:** 7
**Confidence:** 3

**Review:**

This paper presents a thoughtful and methodologically careful framework for ecDNA analysis, highlighting fundamental flaws in existing benchmarks that are often overlooked. The authors’ emphasis on methodological rigor over architectural novelty is well motivated. Each of the three proposed modules is well designed and empirically evaluated, and the work establishes useful baselines for future research in this emerging area. However, I had some additional comments and questions.

1. In Table 1, ECDNA-FORMER using the curated 112 DepMap features only marginally outperforms XGBoost on the same feature set and performs comparably to CytoCellDB without AA* features (312 features). Since ECDNA-FORMER appears to be designed around DepMap-style inputs, it would be useful to clarify whether the model can be evaluated using the larger non-leaky feature set as well.

2. The paper states that ECDNA-FORMER “matches MLP while reducing fold variance by 52%.” However, Table 4 indicates that the MLP baseline achieves higher mean AUROC (0.752 vs. 0.729), while with a higher variance. While reduced variance is valuable, the claim of “matching” performance may be somewhat overstated. This seems somewhat at odds with the paper’s broader narrative that methodological rigor outweighs architectural innovation.

3. A minor issue I wanted to address regarding Table 4 is that the highlighted values in the AUPRC and F1 columns appear to be inconsistent with the reported best-performing methods.

4. The statement that “removing dosage improves performance (+2.4 pp), suggesting overfitting” is somewhat unclear. Are these results based on held-out test data, cross-validation, or training performance? Given that the “no dosage” result is reported as a single split (†), further clarification would help justify the overfitting interpretation.

5. While the paper presents ECLIPSE as a composable pipeline, Section 3.4 notes that the three modules are trained independently and that their joint use is not empirically validated. This makes it 3 separate directions, not one framework and it will be interesting to see how the authors unify them.

6. The external validation of CIRCULARODE is currently limited to two published cell lines. While the results are impressive, broader validation across additional datasets would improve confidence in the generalizability and “zero-shot transfer” claims.

Overall, this is an exciting and carefully executed piece of work that makes important contributions to ecDNA modeling and benchmarking. The emphasis on eliminating leakage, incorporating domain knowledge, and addressing confounding is highly valuable for high-stakes biomedical ML. With clearer positioning of claims, additional validation, and further discussion of limitations, this work has strong potential to serve as a foundational reference in this area.

---

### Meta-Review · Area_Chair_jhzJ · 2026-02-28

**Recommendation:** Accept (Poster)
**Confidence:** 4

**Metareview:**

This paper introduces ECLIPSE, a composable framework for ecDNA prediction, dynamics modeling, and therapeutic vulnerability discovery. Reviewers highlight the paper’s strong emphasis on methodological rigor, including explicit identification of data leakage in existing benchmarks, incorporation of biologically grounded physics constraints into neural SDEs, and causal inference to mitigate confounding in vulnerability discovery. These contributions establish more reliable baselines for computational ecDNA analysis and provide a principled template for high-stakes biomedical ML.

Concerns include limited empirical validation of the fully composable pipeline, uneven generalization across lineages, and relatively limited external validation for the dynamics module. Some claims could be positioned more conservatively.

---

### Decision · Program_Chairs · 2026-03-03

Accept (Poster)